# Dimensional Accuracy of Dental Casting Patterns Fabricated Using Consumer 3D Printers

**DOI:** 10.3390/polym12102244

**Published:** 2020-09-29

**Authors:** Yoshiki Ishida, Daisuke Miura, Taira Miyasaka, Akikazu Shinya

**Affiliations:** 1Department of Life Science Dentistry, School of Life Dentistry at Tokyo, The Nippon Dental University, 1-9-20, Fujimi, Chiyoda-ku, Tokyo 102-8159, Japan; yishida@tky.ndu.ac.jp; 2Department of Dental Materials Science, School of Life Dentistry at Tokyo, The Nippon Dental University, 1-9-20, Fujimi, Chiyoda-ku, Tokyo 102-8159, Japan; daisuke@tky.ndu.ac.jp (D.M.); miyasaka@tky.ndu.ac.jp (T.M.); 3Department of Prosthetic Dentistry and Biomaterials Science, Institute of Dentistry, University of Turku, Lemminkaisenkatu 2, 20520 Turku, Finland

**Keywords:** additive manufacturing, 3D printer, computer-aided-design/computer-aided-manufacturing, dimensional accuracy

## Abstract

Consumer 3D printers have improved considerably due to the evolution of additive manufacturing. This study aimed to examine the accuracy of consumer printers in dental restoration. Cylindrical patterns mimicking a full crown were created and enlarged from 100% to 105% of the original size. Two types of consumer 3D printers, including a fused deposition modeling (FDM) device and a stereolithography (SLA) device, and two types of dental 3D printers, including a multi-jet device and an SLA device, were used to fabricate the patterns. Then, the outer and inner diameters and depths, and surface roughness of the patterns were measured. The changing rates of the outer diameter of models created using dental printers were significantly smaller than those of the models created using consumer printers (*p* < 0.05). Significantly greater surface roughness was obtained in the tooth axis of the model fabricated using the consumer FDM device (*p* < 0.05). However, no significant differences were observed on the surface roughness of both axes between the consumer SLA device and the dental devices (*p* > 0.05). However, FDM showed larger surface roughness than dental devices in both axes (*p* < 0.05). Thus, the SLA consumer printer can be applied to fabricate resin patterns with enlargement editing of 1–3% along the horizontal axis.

## 1. Introduction

3D printers have evolved at a remarkable pace [1]. Consequently, the performance of consumer 3D printers has considerably improved, along with a steady reduction in their price; therefore, they have been widely used to successfully fabricate components for various applications across different fields with high precision [2,3,4,5,6,7]. Until recently, only computer-aided-design (CAD)/computer-aided-manufacturing (CAM) systems have been used to make models for dental restorative applications; this approach involves milling blocks or disks based on a dental model design [5,8]. In the past, closed CAD/CAM systems were commonly used for dental model design, wherein the selection of different CAD or CAM systems cannot be used. Nevertheless, as the Standard Triangle Language (STL) became a standard for the formatting of 3D data, open CAD/CAM design systems were proposed, wherein different CAD or CAM devices can be selected [9,10,11]. Consequently, 3D printers can be used instead of dental CAM devices for dental restoration applications.

In traditional milling devices, because of restrictions posed by the size and applied angle of the milling tool, it is difficult to fabricate complex shapes. In contrast, there are no such limitations with additive manufacturing; thus, 3D printers can be used to fabricate more complex shapes than milling devices [12,13,14]. Owing to the high costs involved, it can take a considerable time before dental 3D printers are widely used for casting dental patterns. However, additive manufacturing can be employed for dental restoration applications using consumer 3D printers, using which dental restoration models with complex designs can be fabricated with uniform quality.

For consumer use, fused deposition modeling (FDM) is the most common type in the category of cheap and affordable category 3D printers. FDM printers lead plastic filament to an extruder head where it is melted and forced out through a small diameter jet onto the surface, where it solidifies. Stereolithography (SLA) is also available in desktop 3D printers. SLA printers use photopolymerization, a process by which a laser beam causes chains of molecules to link together, forming polymers [15]. Consumer FDM printers are used to fabricate 3D models for surgical planning [16,17], whereas consumer SLA printers are used for the fabrication of a dental implant surgical guide [18,19,20]. Compared to SLA, the FDM process is less accurate [21]. However, the use of a consumer FDM 3D printer in fabricating a full denture has been reported. Kim et al. [22] proposed a new digital dental prosthesis, using an FDM printer to make a flask for making a complete denture. Thus, consumer printers are widely applicable to dentistry and could be used to fabricate casting patterns instead of dental printers, due to the increased printing accuracy on consumer printers.

If it is possible to produce casting patterns using consumer 3D printers, digital dentistry can further develop because they would be easier than dental 3D printers to introduce due to the price. However, to the best of our knowledge, the printing accuracy of consumer 3D printers for dental restoration applications has not yet been investigated. Furthermore, if the fabrication accuracy of consumer 3D printers is insufficient for dental restoration applications, improving the printing precision of the consumer 3D printers and that of the dental 3D printers by editing the STL files can address this potential problem. For instance, the STL files can be modified to enlarge the cement space to ensure accurate fabrication. The results might allow us to apply consumer 3D printers to dental restoration instead of dental 3D printers.

The objective of this study was to examine the applicability of consumer 3D printers for fabricating dental restoration models. In particular, cylinder shape casting patterns mimicking a full crown were made using two types of consumer 3D printers. The dimensional accuracy and surface roughness of the fabricated patterns were measured and compared to the results of those fabricated by dental 3D printers. In addition, processing of STL files to improve pattern fitness was investigated. In this study, we hypothesized that it is possible to produce patterns accurately using consumer 3D printers as precise as dental 3D printers.

## 2. Materials and Methods

STL files for a cylinder shape were used to fabricate the full crowns; these data were created using a 3D CAD software (Creo Elements/Direct Modeling Express 4.0 PTC, Needham, MA, USA). The STL files of crowns were designed as follows: 13 mm outer diameter, 10 mm inner diameter, 11 mm outer height, and 10 mm depth, as shown in Figure 1 [4]. The specifications of the two different systems of consumer 3D printers and the two different systems of dental 3D printers used in this study are listed in Table 1.

A fused deposition modeling (FDM) device (MAESTRO, ALT Design, Taipei, Taiwan; hereinafter called MA) and a stereolithography (SLA) device (Nobel 1.0, XYZprinting, New Taipei City, Taiwan; hereinafter called NB) were used as consumer 3D printers for the fabrication of the full crowns. In contrast, a multi-jet device (ProJet DP 3000, 3D Systems, Rock Hill, SC, USA; hereinafter called DP) that discharges the UV curing resin, which is then cured using a UV lamp layer by layer, and an SLA device (DW028D, DWS, Vicenza, Veneto, Italy; hereinafter called DW) were the two dental 3D printers used in our study. The STL files were organized to model the occlusal plane on the fabricating platform to ensure precise margins. Since there were frequently poor fits with patterns made by the consumer printers, enlarged models (101%, 103%, and 105%) of the original design size were also fabricated. Furthermore, 3D printing was performed with the smallest layer thickness in each of the four printers. Removal of the patterns from the platform and their post-processing were completed in accordance with the manufacturer’s instructions for each 3D printer.

The dimensions of the outer and inner diameter and depth of the cylindrical patterns were measured using a microscope (VHX-2000, Keyence, Osaka, Japan). The surface roughness of the outer-side wall of the pattern in the vertical and horizontal directions to the tooth axis was determined using a surface roughness tester (Surfcom 2B, Tokyo Seimitsu, Tokyo, Japan).

The number of repetitions was set to *n* = 6 in all modeling conditions. The changing rate (%) between the six iterations was determined by calculating the percentage change between the observed values and original design values of 100% size for the outer and inner diameters and depth of patterns. For the statistical processing, two-way analysis of variance (ANOVA) was performed for the inner and outer diameters and depth (factor A: printer type, factor B: enlargement ratio) and for the surface roughness (factor A: printer type, factor B: scanning direction). Furthermore, Tukey’s multiple comparisons were carried out for significantly different factors.

## 3. Results

The exterior views are shown in Figure 2a. A raft was created around MA to ensure adhesion to the modeling platform. Some supports were created between the patterns and modeling platform to prevent the occlusal surface from directly touching the platform in NB and DW. On the cervical and ceiling surfaces of MA, imprints of the melted filaments were observed (Figure 2b). In contrast, a grid pattern, which could be attributed to laser scanning, was visible on the cervical and ceiling surfaces of DW (Figure 2c). Moreover, on the marginal surface of MA and DW, laminated layers were observed, but no such layers were found on NB and DP (Figure 2d) [23].

Figure 3 shows the mean and standard deviation values of the changing rates of the outer and inner diameters, and the depth of the patterns fabricated using the consumer and dental 3D printers.

Two-way ANOVA on the changing rates of the outer diameter showed significant differences with factor A (printer type), factor B (enlargement ratio), and their interaction (A × B) (*p* < 0.05). In particular, the resulting outer diameter was small compared to the design value for all types of printers. No significant difference was observed between MA and NB and between DP and DW when the models were fabricated at a size of 100% (*p* > 0.05). Furthermore, there were no significant differences between the MA patterns enlarged to 103%, NB patterns enlarged to 101%, DP patterns at 100%, and DW patterns at 100% (*p* > 0.05).

Two-way ANOVA on the changing rates of the inner diameter showed significant differences with factor A, factor B, and A × B (*p* < 0.05). In particular, the resulting inner diameter was smaller than the design value in the case of all printers at 100%. However, the inner diameter in the case of patterns with an enlargement ratio of 101% for MA and NB, and 100% for DP and DW, had an inner diameter close to the design value.

Two-way ANOVA on the changing rates of depth showed significant differences with factor A, factor B, and A × B (*p* < 0.05). No significant differences were observed for any condition (*p* > 0.05). In particular, the depth values of the fabricated models were close to the design value at 100% enlargement in the case of all printers.

The deviations of the inner and outer diameters and depths of the fabricated patterns from the design values were calculated using the measured values. Significant differences were observed in terms of factor A (printer type). Table 2 shows the deviations (mm) between the measured values of the cylindrical patterns and design values (*n* = 20). In the case of the outer diameter, there were significant differences among the patterns fabricated by all printers (*p* > 0.05), whereas in the case of the inner diameter, no significant difference was observed between the MA and DW patterns and between the DP and DW patterns (*p* > 0.05). Furthermore, in terms of depth, no significant difference was observed among the patterns printed by all printers (*p* > 0.05). Moreover, in the case of the DP patterns, there was no significant difference between the outer and inner diameters (*p* > 0.05). However, the deviations of the outer diameter were significantly larger than those of the inner diameter in the case of models printed by other printers (*p* < 0.05).

Table 3 shows the mean and standard deviation values of the surface roughness of the patterns in the vertical and horizontal directions. Based on the two-way ANOVA on the surface roughness, there were significant differences between factor A (printer type), factor B (scanning direction), and A × B (*p* < 0.05). On the outer surface of the patterns in a direction perpendicular to the modeling platform, the surface roughness was in the following order, MA > NB ≈ DP > DW. In particular, no significant difference was observed between the NB and DP patterns in terms of surface roughness (*p* > 0.05). On the horizontal plane, the surface roughness was in the following order, MA > NB ≈ DP ≈ DW. In particular, no significant difference was observed among the MA, DP, and DW patterns and among the NB, DP, and DW patterns (*p* > 0.05). Furthermore, there was no significant difference between the vertical and horizontal planes in the DW patterns (*p* > 0.05).

## 4. Discussion

After the patent for the FDM device expired, several types of small consumer 3D printers were developed and introduced into the market for personal use [1,2,5,24,25,26]. The development and advancement of these printers took place at a remarkable rate, such that some consumer 3D printers have printing performance comparable to that of industrial 3D printers. The minimum stacking pitches of the dental 3D printer used in this study were approximately 30 µm for the multi-jet and 10 µm for the SLA device. The stacking pitches of the consumer printers used in this study were 50 µm for the FDM device and 25 µm for the SLA device. The pitch of the consumer FDM device was slightly larger than that of the dental 3D printers, but that of the consumer SLA device was almost the same as the multi-jet device. Thus, the consumer 3D printers used in this study had significantly smaller stacking pitches. Therefore, it may be possible to make cylindrical patterns using the consumer 3D printers with almost the same precision as the dental 3D printers.

The dimensional accuracy of cylindrical patterns fabricated using the 3D printers in this study can be ranked as follows, NB > MA > DP = DW. The consumer printers had worse printing accuracy than that of the dental printers.

The printing process of NB is the same as that of DB:SLA using a laser beam to polymerize the resin. It is said that the printing process using a laser or UV beam for curing may have some optical problems, such as monochromatic aberration and astigmatism [15]. Monochromatic aberration occurs for some areas of the build model that lie away from the optical axis. In order to solve these problems, it is necessary to correct the size of the laser spot. Astigmatism occurs when the optical system is unsymmetrical to the optical axis because of a manufacturing error or misalignment of the components. It can be observed even for rays from on-axis object points [15]. The consumer and dental SLA devices were calibrated prior to every printing and were checked to ensure that there was no astigmatism. The NB printing accuracy of outer and inner diameter could have been worse than that of DW due to monochromatic aberration. The correction for the size of the laser spot on NB may be insufficient compared to DW.

The FDM printer used in this study had a heated extruder with 0.4 mm diameter, and a 1.75 mm thermal plasticity filament was extruded from it to proceed to printing layer by layer. The upper layer of an object was printed by pressing against a lower layer of it. This type of 3D printing may demonstrate a curling problem. Curling means that the edge or corners of the parts rise above the part-bed surface. The corners of parts may get thinner in the vertical direction due to a temperature difference between the extruded part and newly added material. Consequently, the surface of the parts is not flat, and the part may move in the part-bed while the printing process is continuing [15,27]. Due to this issue, the printing accuracy of outer and inner diameter on FDM may be worse than that of the dental printers. In addition, it has been reported that the FDM process is inaccurate compared to SLA because the printing accuracy is limited to the extruder size [21,28].

The shrinkage of the printing materials is an unavoidable problem during the printing process and it affects the printing accuracy of most 3D printing technology. In the FDM process, the shrinkage occurs due to thermal contraction when the melted filament solidifies. In the SLA and multi-jet process, the materials also experience shrinkage because of the polymerization process [15]. In this study, the dimension of all specimen produced by four printers showed small values compared to those designed. It indicated that shrinkages occurred when printing the specimens. However, the percentage of the shrinkage was different among the printers, and the changing rates of consumer printers were larger than those of dental printers. Our results suggest that the compensation for shrinkage may not be adequate in the consumer printers in the horizontal direction. In contrast, there was no significant difference in depth. This suggests that the consumer printers can fabricate cylindrical patterns as accurately as the dental printers in the vertical direction.

Ensuring smooth surfaces is also an important factor in the dental field [27]. The surface roughness in the vertical direction was ranked in the following order, MA > NB > DP > DW, whereas that in the horizontal direction was ranked in the following order, MA > DP = DW > NB. In particular, the vertical roughness was affected by the stacking pitch; the smallest surface roughness on the vertical plane was observed in the DW patterns, which had the smallest stacking pitch. In contrast, the horizontal roughness may be affected by the type of printing material used. No significant difference was observed among DP, DW, and NB patterns in regard to horizontal surface roughness. However, Hambali et al. [29] reported that the surface roughness of the samples fabricated by the FDM printer was improved by approximately 92% via immersion in acetone solution for 300 s. It may be possible to obtain a smooth surface on the cylindrical patterns fabricated by the FDM device by applying the aforementioned treatment. However, the effect on the dimension change is not discussed in the report, and thus further investigation is needed.

In this study, consumer 3D printers were used to fabricate casting patterns. Our experimental data suggested that the printing accuracy of the consumer printers was as precise as the dental printers in the vertical direction. However, in the horizontal direction, the patterns fabricated using consumer printers showed shrinkage, and thus enlargement editing of 1–3% for the STLs is necessary in MA and NB to accurately make patterns. For the surface roughness, patterns fabricated by the consumer SLA printer were as smooth as those of dental printers, but additional treatment is required to increase smoothness for the consumer FDM printer.

The main limitation of this study was the design of the cylindrical patterns fabricated by 3D printers, which was simplified to a full crown to perform precise measurements. An anatomically-shaped crown is more complex than the design used in this study, and it is challenging to clarify the printing accuracy with the method used in this study. Thus, it is necessary to develop other methods for measuring the dimension of the complex shape accurately so that further studies are required to fabricate casting patterns using consumer 3D printers.

## 5. Conclusions

Cylindrical patterns were fabricated using a consumer FDM 3D printer and a consumer SLA 3D printer, and the printing accuracy of the fabricated patterns was compared with those obtained using dental printers. Our results showed that the dimensional accuracy of cylindrical patterns fabricated using the 3D printers in this study can be ranked as follows, NB > MA > DP = DW. The consumer printers had worse printing accuracy than that of the dental printers. An enlargement adjustment of 1–3% along the horizontal axis was necessary to realize the set design value for consumer printers. In terms of surface roughness, the consumer SLA printer could fabricate patterns as smooth as those fabricated using dental printers. In contrast, patterns fabricated by the consumer FDM printer were significantly larger than the dental printers. Thus, the results of our study indicated that consumer SLA printers have the potential to be used for the fabrication of patterns for applications, such as dental restoration in the field of dentistry.

## Figures and Tables

**Figure 1 polymers-12-02244-f001:**
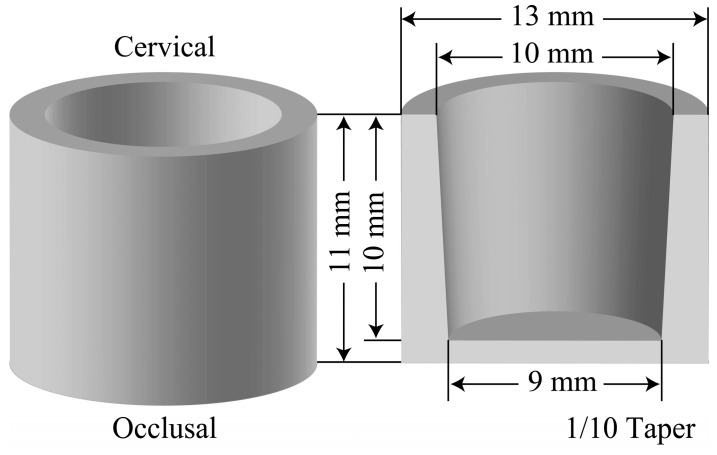
Dimensions of the cylindrical patterns of the model crowns.

**Figure 2 polymers-12-02244-f002:**
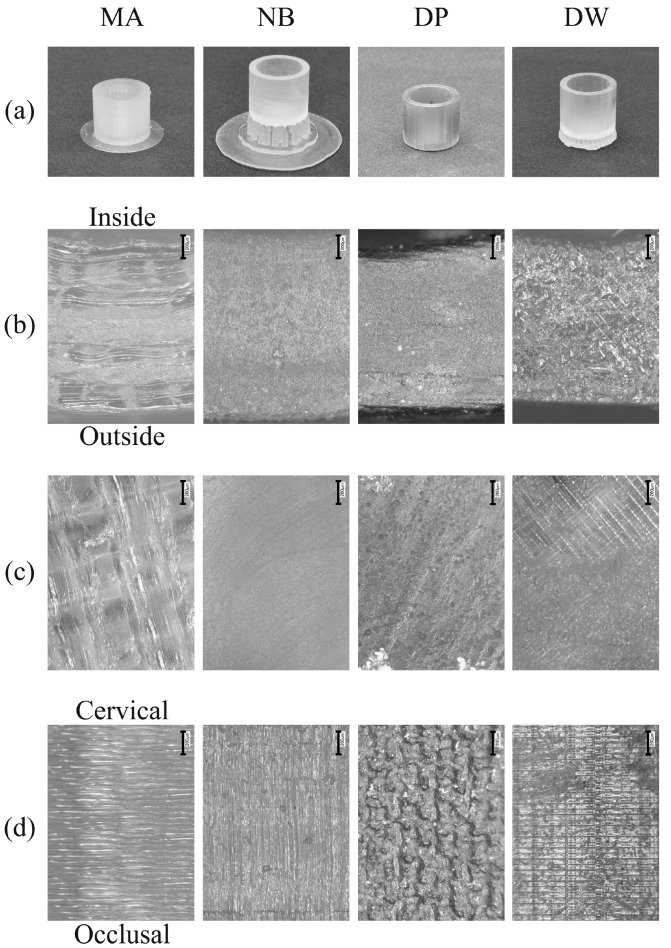
Images of the crowns fabricated using 3D printers (200× magnification, “I” presents 200 μm). (**a**) Exterior view; (**b**) basal plane of the cervical part; (**c**) inner upper surface; (**d**) outer sidewall of the patterns. The scale bar is 200 μm.

**Figure 3 polymers-12-02244-f003:**
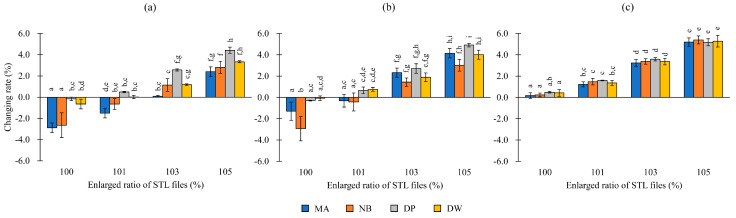
Changing rates (%) of patterns fabricated by each 3D printer. (**a**) Outer diameter; (**b**) inner diameter; (**c**) depth. The superscripts with the same letters indicate combinations that are not significantly different (*p* > 0.05).

**Table 1 polymers-12-02244-t001:** The 3D printers and printing materials used in this study.

Code	Product Name	System	Printing Materials (Lot No.)	Manufacture	Minimum Stacking Pitch (µm)
MA	MAESTRO	Fused deposition Modeling	PLA filament natural (X000C36R1B)	ALT Design (Taipei, Taiwan)	50
NB	Nobel 1.0	Stereolithography	Photopolymer Resin (clear) (RUGNR628GB3EM73W0144)	XYZ Printing (New Taipei City, Taiwan)	25
DP	ProJetDP3000	Multi-jet	VisiJet DP200 (DP132502A)	3D Systems (Rock Hill, SC, USA)	29–32
DW	DW028D	Stereolithography	RF080 (4120225)	DWS (Veneto, Italy)	10

**Table 2 polymers-12-02244-t002:** The deviation (mm) from the design value of the inner and outer diameters and the depth of fabricated patterns.

Printer	Outer Diameter		Inner Diameter		Depth	
MA	−0.35	(0.06)	−0.10	(0.06) ^a,c^	0.03	(0.03) ^e^
NB	−0.25	(0.10)	−0.18	(0.08) ^d^	0.04	(0.03) ^e^
DP	−0.05	(0.03) ^a,b^	−0.03	(0.03) ^b,e^	0.05	(0.04) ^e^
DW	−0.17	(0.07) ^c,d^	−0.06	(0.05) ^a,b^	0.04	(0.02) ^e^

The superscripts with the same letters indicate combinations that are not significantly different. (*p* > 0.05).

**Table 3 polymers-12-02244-t003:** Surface roughness (µmRa) on the sidewall of patterns measured in the vertical and horizontal directions to the tooth axis.

Printer	Vertical Direction		Horizontal Direction	
MA	7.33	(0.91)	3.44	(0.71) ^b,d^
NB	5.83	(0.92) ^a^	1.90	(0.34) ^c^
DP	4.61	(0.49) ^a,b^	2.09	(0.14) ^c,d^
DW	1.39	(0.10) ^c^	2.73	(0.13) ^c,d^

The superscripts with the same letters indicate combinations that are not significantly different. (*p* > 0.05).

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
