# Peer review of "Dimensional Accuracy of Dental Casting Patterns Fabricated Using Consumer 3D Printers"

_polymers, 2020, doi:10.3390/polym12102244_

Round 1
Reviewer 1 Report
Dear Authors
Please add below references in your introduction part;
a) https://www.sciencedirect.com/science/article/pii/B9780081024768000025
b) https://www.sciencedirect.com/science/article/pii/B9780081024768000190
c) https://www.sciencedirect.com/science/article/pii/B9780081024768000244
This above information very useful in between line number 32- 70.
I discussion add information from this paper and related with your reported work.
https://www.thieme-connect.com/products/ejournals/abstract/10.1055/s-0040-1705243
Author Response
Comments for reviewer 1
Dear Authors
Please add below references in your introduction part;
- a) https://www.sciencedirect.com/science/article/pii/B9780081024768000025
- b) https://www.sciencedirect.com/science/article/pii/B9780081024768000190
- c) https://www.sciencedirect.com/science/article/pii/B9780081024768000244
This above information very useful in between line number 32- 70.
I discussion add information from this paper and related with your reported work.
https://www.thieme-connect.com/products/ejournals/abstract/10.1055/s-0040-1705243
Response
Thank you very much for the suggestions. The references were added in the manuscript, Ref number is 6 and 7.
Reviewer 2 Report
The manuscript is presenting an interesting topic on 3D printing of a Cylindrical resin polymer pattern using SLA/FDM and is defining a method to boost the accuracy of the printing. the surface roughness of the parts was compared.
the comments and questions below are essential to take in before we proceed with this manuscript:
-from Table 1, it was not clearly stated what causes the significantly larger diameter value for the inner diameter on the printed parts from different printers. please clearly state this and also add in the conclusion.
-SEM images and the microstructures presented must be cited to DOI: 10.1016/j.matchar.2020.110167
-the values at table 2 could be presented in a graph. this eases the reading and comparision.
-SEM images: in case C-DW, the half of the part shows a different morphology, how this happens?
-this is odd to me that the In SLA and multi-jet 200 process, the materials shrink too. what cause this? i don't think this compensates as the diameter difference is not much in these cases.
when authors take my comments in precisely and answer to my questions i can reconsider in my decision.
Author Response
Comments for reviewer 2
The manuscript is presenting an interesting topic on 3D printing of a Cylindrical resin polymer pattern using SLA/FDM and is defining a method to boost the accuracy of the printing. the surface roughness of the parts was compared.
Response
Thank you very much for the suggestions. Revisions were made accordingly.
the comments and questions below are essential to take in before we proceed with this manuscript:
-from Table 1, it was not clearly stated what causes the significantly larger diameter value for the inner diameter on the printed parts from different printers. please clearly state this and also add in the conclusion.
Response
The revision was made in the discussion and the conclusion section. Also, Table 1 was modified.
-SEM images and the microstructures presented must be cited to DOI: 10.1016/j.matchar.2020.110167
Response
The report was added to the manuscript as a new reference in Ref No 23.
-the values at table 2 could be presented in a graph. this eases the reading and comparision.
Response
Table 2 was replaced with a graph.
-SEM images: in case C-DW, the half of the part shows a different morphology, how this happens?
Response
The morphology in the lower half of Figure 2c (DW) was because of post-treatment with an unpolymerized monomer. However, authors have checked that such an area does not affect the results of the depth.
-this is odd to me that the In SLA and multi-jet 200 process, the materials shrink too. what cause this? i don't think this compensates as the diameter difference is not much in these cases.
Response
It might be because of Polymerization shrinkage. Photopolymer materials were used in SLA and multi-jet process. There was shrinkage when the materials were cured. This phenomenon is also explained in the paper (http://dx.doi.org/10.14748/ssmd.v1i1.1647). This paper is cited in our manuscript as reference number 15. The dental printers were used as controls because they can fabricate samples accurately. However, slightly small differences between the value observed and designed was also found in the dental printers, so that more investigations will be needed to clarify them.
when authors take my comments in precisely and answer to my questions i can reconsider in my decision.
Response
We appreciate your comments and revised the manuscript accordingly.
Reviewer 3 Report
This manuscript untitled “Dimensional accuracy of resin patterns fabricated 2 using consumer 3D printers”. Aim of this paper is quite interesting, and is in the scope of this journal. Generally, there are grammatical errors in this manuscript. It is recommended that it would be revised again by English scientific writer.
For these, there are numerous issues in the present manuscript that need to be addressed before publication:
Title: The title is not informative enough, I think is important to have the word “dental” or similar in the title.
Abstrat:
- The Aim must be clear e Is not it clear what the study found.
- In the results, is important to show more information, add some the p-values.
Introduction
- In the first sentence, please modified and add a reference.
- What is the novelty of this paper? And what was the importance of this study? Please clarify in this appropriate section.
- Page 2 line 58 - The reference is on final of the sentence. But, the authors et al. in the text of the manuscript, references should come immediately afterwards.
- What was your hypothesis null hypothesis?
Materials and results:
- Tables 1- mention the manufacturer and city/ country. Also insert a column for the lot and expiration date of the materials used in this printer.
- How was the sample size calculated? How was the sample calculated? Did authors performed power analysis to evaluate if this sample size was appropriate?
- Improve the quality of all images, has little resolution.
(Discussion)
- You should conduct a discussion with the literature regarding the results you found.
- Please, identified what was the limitation of this study? And also, future perspectives for correct these limitations.
- The authors can used these sentences “The dimensional accuracy of cylindrical patterns fabricated using the 3D printers in this study can be ranked as follows: NB>MA>DP=DW. The consumer printers had worse printing accuracy than that of the dental printers” for the abstract and the conclusion.
(conclusions)
- The conclusions should be adjusted. Please, need to format this part of the manuscript. The conclusions should be more synthetic, and should aim at your goals
References
- Check reference’s format in the manuscript, and in the references. The titles of references have different format (reference 25) and identification of the name are also in different formats.
Author Response
Comments for reviewer 3
This manuscript untitled “Dimensional accuracy of resin patterns fabricated 2 using consumer 3D printers”. Aim of this paper is quite interesting, and is in the scope of this journal. Generally, there are grammatical errors in this manuscript. It is recommended that it would be revised again by English scientific writer.
For these, there are numerous issues in the present manuscript that need to be addressed before publication:
Response
Thank you very much for the suggestions. Revisions were made accordingly.
Title: The title is not informative enough, I think is important to have the word “dental” or similar in the title.
Response
The title was revised as “Dimensional accuracy of dental resin casting patterns fabricated using consumer 3D printers” based on reviewer’s advice.
Abstrat:
The Aim must be clear e Is not it clear what the study found.
In the results, is important to show more information, add some the p-values.
Response
The revision was made in the abstract. Please see and confirm new sentence.
Introduction
In the first sentence, please modified and add a reference.
Response
A reference was added for the first sentence with Ref. 1 (Attaran, M. The rise of 3-D printing: The advantages of additive manufacturing over traditional manufacturing. Business Horizons 2017, 60, 677–688, doi:10.1016/j.bushor.2017.05.011.)
What is the novelty of this paper? And what was the importance of this study? Please clarify in this appropriate section.
Response
The revision was made in the introduce section.
“If it is possible to produce casting patterns using consumer 3D printers, digital dentistry can further develop because they would be easier than dental 3D printers to introduce due to the price.” was added at the beginning of 3rd paragraph, and “The results might allow us to apply consumer 3D printers to dental restoration instead of dental 3D printers.” was added end of 3rd paragraph.
Page 2 line 58 - The reference is on final of the sentence. But, the authors et al. in the text of the manuscript, references should come immediately afterwards.
Response
The correction was made in the manuscript. Thanks for your kind advices.
What was your hypothesis null hypothesis?
Response
The hypothesis was added in the introduction section. “In this study, we hypothesized that it is possible to produce patterns accurately using consumer 3D printers as precise as dental 3D printers.” was added in the end of introduction.
Materials and results:
Tables 1- mention the manufacturer and city/ country. Also insert a column for the lot and expiration date of the materials used in this printer.
Response
The manufacturer and city/country were added in Table 1. Unfortunately, the expiration date could not be found for these materials. However, this experiment was performed within a month after materials had arrived.
How was the sample size calculated? How was the sample calculated? Did authors performed power analysis to evaluate if this sample size was appropriate?
Response
Power analysis was performed, and the number of repetitions calculated was 4.6. In this paper, the number was set to n = 6, so it should have been large enough.
Improve the quality of all images, has little resolution.
Response
The images were replaced with high resolution.
(Discussion)
You should conduct a discussion with the literature regarding the results you found.
Please, identified what was the limitation of this study? And also, future perspectives for correct these limitations.
Response
Revision was made in the discussion section about the limitation of this study.
The authors can used these sentences “The dimensional accuracy of cylindrical patterns fabricated using the 3D printers in this study can be ranked as follows: NB>MA>DP=DW. The consumer printers had worse printing accuracy than that of the dental printers” for the abstract and the conclusion.
Response
The sentences were added in the text based on the reviewer’s advice. Thanks for valuable comment.
(conclusions)
The conclusions should be adjusted. Please, need to format this part of the manuscript. The conclusions should be more synthetic, and should aim at your goals
Response
Revision was made in the conclusion section.
References
Check reference’s format in the manuscript, and in the references. The titles of references have different format (reference 25) and identification of the name are also in different formats.
Response
Correction was made in the references.
Round 2
Reviewer 3 Report
This research is under the scope of this journal; the topic is relevant for readers and this research deals with potentially significant knowledge to the field and an open new way for future studies. Aim of this paper is quite interesting.
The authors improved the quality of the manuscript.